# Characteristics and Outcomes of COVID-19-Related Hospitalization among PLWH

**DOI:** 10.3390/jcm11061546

**Published:** 2022-03-11

**Authors:** Roberta Gagliardini, Alessandra Vergori, Patrizia Lorenzini, Stefania Cicalini, Carmela Pinnetti, Valentina Mazzotta, Annalisa Mondi, Ilaria Mastrorosa, Marta Camici, Simone Lanini, Marisa Fusto, Jessica Paulicelli, Maria Maddalena Plazzi, Luisa Marchioni, Chiara Agrati, Anna Rosa Garbuglia, Pierluca Piselli, Emanuele Nicastri, Fabrizio Taglietti, Fabrizio Palmieri, Gianpiero D’Offizi, Enrico Girardi, Francesco Vaia, Andrea Antinori

**Affiliations:** National Institute for Infectious Diseases Lazzaro Spallanzani, IRCCS, Via Portuense, 292, 00149 Roma, Italy; roberta.gagliardini@inmi.it (R.G.); patrizia.lorenzini@inmi.it (P.L.); stefania.cicalini@inmi.it (S.C.); carmela.pinnetti@inmi.it (C.P.); valentina.mazzotta@inmi.it (V.M.); annalisa.mondi@inmi.it (A.M.); ilaria.mastrorosa@inmi.it (I.M.); marta.camici@inmi.it (M.C.); simone.lanini@inmi.it (S.L.); marisa.fusto@inmi.it (M.F.); jessica.paulicelli@inmi.it (J.P.); maria.plazzi@inmi.it (M.M.P.); luisa.marchioni@inmi.it (L.M.); chiara.agrati@inmi.it (C.A.); annarosa.garbuglia@inmi.it (A.R.G.); pierluca.piselli@inmi.it (P.P.); emanuele.nicastri@inmi.it (E.N.); fabrizio.taglietti@inmi.it (F.T.); fabrizio.palmieri@inmi.it (F.P.); gianpiero.doffizi@inmi.it (G.D.); enrico.girardi@inmi.it (E.G.); francesco.vaia@inmi.it (F.V.); andrea.antinori@inmi.it (A.A.)

**Keywords:** HIV, COVID-19, SARS-CoV-2

## Abstract

Background: There is conflicting evidence for how HIV influences COVID-19 infection. The aim of this study was to compare characteristics at presentation and the clinical outcomes of people living with HIV (PLWH) versus HIV-negative patients (non-PLWH) hospitalized with COVID-19. Methods: Primary endpoint: time until invasive ventilation/death. Secondary endpoints: time until ventilation/death, time until symptoms resolution. Results: A total of 1647 hospitalized patients were included (43 (2.6%) PLWH, 1604 non-PLWH). PLWH were younger (55 vs. 61 years) and less likely to be with PaO_2_/FiO_2_ < 300 mmHg compared with non-PLWH. Among PLWH, nadir of CD4 was 185 (75–322) cells/μL; CD4 at COVID-19 diagnosis was 272 cells/μL (127–468) and 77% of these were virologically suppressed. The cumulative probability of invasive mechanical ventilation/death at day 15 was 4.7% (95%CI 1.2–17.3) in PLWH versus 18.9% (16.9–21.1) in non-PLWH (*p* = 0.023). The cumulative probability of non-invasive/invasive ventilation/death at day 15 was 20.9% (11.5–36.4) in PLWH versus 37.6% (35.1–40.2) in non-PLWH (*p* = 0.044). The adjusted hazard ratio (aHR) of invasive mechanical ventilation/death of PLWH was 0.49 (95% CI 0.12–1.96, *p* = 0.310) versus non-PLWH; similarly, aHR of non-invasive/invasive ventilation/death of PLWH was 1.03 (95% CI 0.53–2.00, *p* = 0.926). Conclusion: A less-severe presentation of COVID-19 at hospitalization was observed in PLWH compared to non-PLWH; no difference in clinical outcomes could be detected.

## 1. Introduction

COVID-19 disease, related to severe acute respiratory syndrome coronavirus 2 (SARS-CoV-2), emerged as the greatest health issue of 2020.

To date, no conclusive evidence about the relationship between COVID-19 and HIV infection is available. In fact, many small cohort studies failed to demonstrate a higher risk rate of COVID-19 or more aggressive course of disease in people living with HIV (PLWH) compared to those without HIV (non-PLWH) [1,2,3,4,5,6,7,8,9].

However, larger and more recent studies reported a higher risk of poor outcomes of COVID-19 in PLWH, in particular a higher mortality rate [10,11,12,13,14], higher rate of hospitalization due to COVID-19 [15] or higher rate of in-hospital intubation [11]. 

The WHO Global Clinical Platform for COVID-19 affirmed that HIV appeared to be a significant independent risk factor for severe or critical illness at hospital admission and in-hospital mortality. In this analysis, including more than 15,000 PLWH, data were predominantly from South Africa, and this may limit the generalizability of the results [16].

Two small studies from the UK and France suggested a substantial morbidity or mortality from COVID-19 among Black PLWH [16,17]. Moreover, PLWH might have high prevalence of risk factors for severe SARS-CoV-2 infection, including hypertension, diabetes, cardiovascular disease, lung disease, smoking, and the male sex [17]. Uncontrolled HIV infection or advanced immunodeficiency, in the context of additional co-morbidities, might have a higher risk of COVID-19-related death [18]. 

Moreover, limited information about the difference in levels of inflammatory and immune markers among people hospitalized with COVID-19 by HIV serostatus is available [10,15].

Thus, continued data collections that explores the coinfection between PLWH and COVID-19 are warranted to clarify their mutual relationship and the role of traditional risk factors for COVID-19 outcomes in PLWH.

The aim of this study was to compare characteristics at presentation and clinical outcomes of PLWH versus HIV-negative patients hospitalized with COVID-19.

## 2. Materials and Methods

We conducted a retrospective cohort study in the INMI COVID-19 Database of L. Spallanzani Institute in Rome (Italy), which contains data from patients who had a diagnosis of COVID-19 from January 2020 and were hospitalized in L. Spallanzani Institute in Rome. INMI COVID-19 Database was approved by the local INMI, Rome Ethical Committee, and patients provided written informed consent. The study was performed in accordance with the Declaration of Helsinki. INMI COVID-19 Database retrieves epidemiological, demographic, clinical and laboratory data of patients, as well therapy prescribed for COVID-19 patients.

Patients were included in this study if the following inclusion criteria were satisfied: ≥18 years of age, a diagnosis of SARS-CoV-2 infection, defined as positive RT-PCR from nasal/oropharyngeal (NP/OP) swab; admitted at INMI L. Spallanzani Institute in Rome (Italy). 

Pneumonia was evaluated by CT scan. All patients had PaO_2_/FiO_2_ ratio measured at admission by arterial blood gas analysis.

The start of follow-up (baseline) for each patient was the time of admission to the hospital. All patients were followed up from baseline until death, discharge, last available visit or the administrative censored date of 1 June 2021, whichever occurred first.

Primary endpoint of this study was the evaluation of time from baseline to invasive ventilation or death (whichever occurred first). 

Secondary endpoints were: (i) the evaluation of time from start of therapy to non-invasive or invasive ventilation or death (whichever occurred first), (ii) time until symptoms resolution, defined as resolution of fever (first day without fever) or weaning from oxygen (first day without oxygen supplementation). Non-invasive ventilation includes CPAP and NIV.

Baseline demographic characteristics, symptoms, immune and inflammatory marker levels were compared by HIV serostatus. 

Use of therapy with immuno-modulants (e.g., anti-IL6, anti-JAK), corticosteroids, heparin, hydroxychloroquine or lopinavir/ritonavir was based on local medical consensus, guidelines and the clinicians’ own opinions.

For the statistical analysis, chi-square or Wilcoxon rank sum (Mann–Whitney) tests were used to compare categorical or continuous variables in descriptive analyses, respectively. 

Kaplan–Meier survival method was used to estimate the cumulative proportion of patients experiencing study endpoints, with a corresponding 95% confidence interval (CI); differences between PLWH versus HIV-negative patients (non-PLWH) were evaluated by the log-rank test. 

In order to control for measured confounders, a Cox regression model was used, adjusted for the following variables: age, gender, comorbidities, ratio of arterial oxygen partial pressure to fractional inspired oxygen (PaO_2_/FiO_2_) and pneumonia at admission to the hospital.

A sensitivity analysis including only patients with COVID-19 pneumonia was performed. 

All statistical analyses were performed with STATA.

## 3. Results

### 3.1. Population Characteristics 

A total of 1647 patients hospitalized with a diagnosis of COVID-19 were evaluated in this analysis, of whom 43 were PLWH and 1604 were non-PLWH. Of 1647 patients, 33.4% were female, the median age was 61 years (IQR 50–74), 82.8% were of Italian nationality, and 72.4% had at least one comorbidity. At hospitalization, 90% had a diagnosis of pneumonia, median baseline PaO_2_/FiO_2_ 321 (IQR 230–386) mmHg, days from onset of symptoms to hospitalization 7 (IQR 4–10). Demographic characteristics, signs, symptoms and biomarkers are shown in Table 1.

PLWH were significantly younger and less likely to be with PaO_2_/FiO_2_ below 300 mmHg at admission, with less alterations in D-dimer, ferritin and slight alternations in lymphocytes compared with non-PLWH (Table 1). 

Symptoms at presentation were similar in the two groups, apart from headache, which was more frequent in PLWH (Table 1).

Viro-immunological characteristics of the PLWH group were the following: median CD4+ cells count at COVID-19 diagnosis: 272 cells/μL (IQR127–468); median CD4+ cells count at lowest point: 185 cells/μL (IQR75–322); 32% with a previous diagnosis of AIDS, all had antiretroviral therapy; and 77% had HIV-1 RNA < 50 copies/mL at COVID-19 diagnosis. Characteristics of PLWH who were not virologically suppressed are shown in detail in Table 2, but a sub-analysis of the above-specified endpoints was not performed in this population due to the small dimension of this subgroup.

### 3.2. Invasive Ventilation or Death

Overall, 306 patients over 1647 (18.6%) underwent to invasive ventilation or death, two in PLWH group.

By Kaplan–Meyer analysis, the estimated probability of invasive ventilation or death was 4.7% (95% CI 1.2–17.3) in PLWH and 18.9% (95% CI 16.1–21.1) in non-PLWH 15 days after hospitalization (log-rank test = 0.023).

Following the adjustment for age, gender, comorbidities, PaO_2_/FiO_2_ and pneumonia at admission, the adjusted hazard ratio (aHR) of mechanical ventilation/death of PLWH was 0.49 (95% CI 0.12–1.96, *p* = 0.310) versus non-PLWH.

### 3.3. Non-Invasive or Invasive Ventilation or Death

The cumulative probability of non-invasive or invasive ventilation/death at day 15 was 20.9% (11.5–36.4) in PLWH versus 37.6% (35.1–40.2) in non-PLWH (log-rank test = 0.044). 

Following the adjustment for the aforementioned variables, the aHR of non-invasive or invasive ventilation/death of PLWH was 1.03 (95% CI 0.53–2.00, *p* = 0.926). 

### 3.4. Symptoms Resolution 

The cumulative probability of symptoms resolution at day 15 was 65.3% (48.4–81.7) in PLWH versus 50.5% (47.6–53.4) in non-PLWH (log-rank test = 0.041).

In the adjusted model, the probability of symptoms resolution at day 15 PLWH was similar in the two groups (aHR 0.92; 0.63–1.36; *p* = 0.691). 

### 3.5. Sensitivity Analysis

A total of 1494 patients hospitalized with diagnoses of COVID-19 pneumonia were evaluated in this sensitivity analysis. By Kaplan–Meyer analysis, the estimated probabilities of invasive ventilation or death was 6.5% (95% CI 1.7–23.4) in PLWH and 20.5% (95% CI 18.4–23.4) in non-PLWH after 15 days of hospitalization (log-rank test = 0.073).

Following adjustment for age, gender, comorbidities, PaO_2_/FiO_2_, aHR of mechanical ventilation/death of PLWH was 0.46 (95% CI 0.11–1.85, *p* = 0.272) versus non-PLWH.

The cumulative probability of non-invasive or invasive ventilation/death at day 15 was 29.0% (16.3–48.4) in PLWH versus 40.69% (38.3–43.7) in non-PLWH (log-rank test = 0.276). Following adjustment for the same variables, aHR of non-invasive or invasive ventilation/death of PLWH was 1.02 (95% CI 0.53–1.98, *p* = 0.954). 

The cumulative probability of symptoms resolution at day 15 was 58.2% (39.8–77.7) in PLWH versus 49.9% (39.8–77.7) in non-PLWH (log-rank test = 0.162). 

## 4. Discussion

The impact of HIV coinfection on the clinical course of patients with COVID-19 has yet to be fully clarified.

Our findings suggest that HIV status did not significantly impact clinical outcomes in patients with SARS-CoV-2 coinfection and are in line with other large studies conducted in the US, which evaluated clinical evolution and mortality among PLWH and COVID-19 [19,20]. This study was conducted on a cohort of 7576 veterans (Veterans Aging Cohort Study, VACS) and states that people with HIV are more likely to be assessed for COVID-19, but there is no difference in the risk of hospitalization, ICU admission, intubation or death between HIV-positive and HIV-negative patients [19].

A previous study investigating clinical outcomes in a small case series of patients admitted to our institute, did not find an increased risk and severity of COVID-19 in PLWH despite a significant T-cell activation and inflammatory profile, suggesting a potential role of HIV-driven immunological dysregulation in avoiding immune-pathogenetic processes [21]. 

Moreover, lymphopenia in COVID-19 infection seems to be a relevant factor for the progression of severity of illness [22]. 

CD4 lymphopenia was reported with variable severity of COVID-19 infection. It was hypothesized that a low CD4 count might protect PLWH from the development of the cytokine storm, which is part of the COVID-19 clinical syndrome, and potentially reduce some of the severe manifestations of COVID-19 infection [23] resulting in a protective feature in preventing severe clinical manifestations of COVID-19 infection [6,7,24,25]. In our cohort, PLWH had quite low CD4+ levels at COVID-19 diagnosis (272 CD4+ cells/μL [IQR 127–468]).

A study derived from a large US database included more than 50,000 patients with COVID-19, of whom about 49,000 were HIV-negative and 400 were HIV-positive, mostly of African American ethnicity, obese, hypertensive, with diabetes, renal failure and other concomitant diseases. This demonstrated that the severity of COVID-19 in PLWH might be dependent on the presence of comorbidities [9]. Subsequent to this study, Boulle and colleagues showed an approximately 2.7-fold increased risk of death from COVID-19 in the HIV-positive population compared to the HIV-negative population, after adjustment for confounding factors such as age, the presence of other concomitant infectious and non-infectious diseases [26]. Both studies emphasize that the severity of COVID-19 in HIV-positive patients was not due to the presence of HIV itself, but to the ageing and comorbidities of these individuals [9,26].

Moreover, a systematic review by Cooper T. et al. on 70 PLWH from 8 studies did show that well-controlled HIV patients were not associated with poorer outcomes of COVID-19 infection than the general population, which was similar to our observation [27].

According to the WHO Global Clinical Platform for COVID-19, HIV infection is an independent risk factor of worse COVID-19 outcomes, but these data were predominantly from South Africa, mainly without information on antiretroviral therapy. Moreover, not all potentially relevant risk factors, such as body mass index, were evaluated [16].

In our cohort, HIV-positive patients were significantly younger than the HIV-negative comparator, mostly virologically suppressed before COVID-19 and with a median CD4 count of 272 cells/μL; hence, these findings may not be applicable to a population with poorly controlled HIV or AIDS. The two groups did not differ in terms of COVID-19 symptoms (only headache was more frequent in PLWH) and comorbidities, with the exception of hypertension and cardiovascular diseases found in a significantly higher proportion of HIV-negative patients.

At time of admission to hospital, hypoxemia, the need for oxygen supplementation, and pneumonia were significantly more frequent in those without HIV; these differences could be explained by the older age of those without HIV and/or increased prevalence of cardiovascular diseases.

Among the seven PLWH not virologically suppressed, only two PLWH had an AIDS diagnosis at admission; the other patients had more than one comorbidity. All of them were discharged alive with no COVID-related complications, except one patient who required invasive mechanical ventilation and died for reasons not related to COVID-19. 

The main limitation of this study is the retrospective analysis, performed in a single medical center; therefore, results may not be generalizable to other centers and may also be limited by a small sample size. Larger prospective studies from multiple centers will be needed to verify findings from this study and to examine the impact of poorly controlled HIV on the clinical course of SARS-CoV-2. Additionally, given the lack of immunological data on all study subjects, we are unable to report on the immunological response of PLWH to SARS-CoV-2 infection as compared with patients with COVID-19 but not with HIV infection. Taken together, our findings are reassuring in that HIV-positive patients on antiretroviral therapy may not experience significantly worse outcomes in SARS-CoV-2, even though these data should be considered with caution. This urges us not to rush to conclusions about a reversal of previous data on large cohorts telling us that HIV still remains a risk factor for COVID-19.

## Figures and Tables

**Table 1 jcm-11-01546-t001:** Demographics, comorbidities, baseline characteristics, presenting symptoms and biomarkers of the overall population and of the 2 groups (non-PLWH and PLWH).

Characteristics	Total Population	Non-PLWH	PLWH	*p*-Value
	N = 1647	N = 1604	N = 43	
Gender				
Male, *n*(%)	1097 (66.6)	1065 (66.4)	32 (74.4)	0.271
Age, years, median (IQR)	61 (50–74)	61 (50–74)	55 (47–62)	0.003
Not Italian, *n* (%)	283 (17.2)	268 (16.7)	15 (34.9)	0.002
BMI, median (IQR)	25.9 (24.0–29.4)	26.0 (24.0–29.4)	24.7 (22.0–28.4)	0.062
At least 1 comorbidity, *n* (%)	1192 (72.4)	1165 (72.6)	27 (62.8)	0.154
Obesity, *n* (%)	226 (20.9)	223 (21.1)	3 (10.3)	0.255
Diabetes, *n* (%)	273 (16.6)	270 (16.8)	3 (7.0)	0.086
Heart diseases, *n* (%)	422 (25.6)	419 (26.1)	3 (7.0)	0.005
Hypertension, *n* (%)	644 (39.1)	638 (39.8)	6 (14.0)	0.001
Chronic lung diseases, *n* (%)	241 (14.6)	237 (14.8)	4 (9.3)	0.316
CNS diseases, *n* (%)	178 (10.8)	174 (10.9)	4 (9.3)	0.723
Therapies				
HCQ, *n* (%)	403 (24.5)	398 (24.8)	5 (11.6)	0.047
PI/b, *n* (%)	455 (27.6)	438 (27.3)	17 (39.5)	0.077
Remdesivir, *n* (%)	454 (27.6)	438 (27.3)	16 (37.2)	0.152
Immune therapy, *n* (%)	158 (9.6)	156 (9.7)	2 (4.7)	0.265
LMWH, *n* (%)	1316 (79.9)	1284 (80.1)	32 (74.4)	0.363
Steroids, *n* (%)	956 (58.0)	937 (58.4)	19 (44.2)	0.062
PaO_2_/FiO_2_ ratio <300 at admission, *n* (%)	627 (38.1)	619 (38.6)	8 (18.6)	0.011
PaO_2_/FiO_2_ ratio at admission, median (IQR)	321 (230–386)	319 (229–386)	376 (314–438)	0.007
Worst PaO_2_/FiO_2_ ratio during FU, median (IQR)	234 (138–350)	232 (138–348)	338 (232–414)	0.001
Pneumonia, *n* (%)	1494 (90.7)	1463 (91.2)	31 (72.1)	<0.001
Oxygen supplementation, *n* (%)	1244 (75.6)	1217 (75.9)	27 (62.8)	0.048
Venturi Mask, *n* (%)	1077 (65.4)	1053 (65.7)	24 (55.8)	0.181
NIV, *n* (%)	475 (28.8)	467 (29.1)	8 (16.6)	0.133
OTI, *n* (%)	234 (14.2)	232 (14.5)	2 (4.7)	0.069
Time from symptoms to admission, days, median (IQR)	7 (4–10)	7 (4–10)	8 (5–10)	0.148
Time until viral clearance from symptoms, median (IQR)	19 (13–27)	19 (13–27)	21 (14–32)	0.252
Headache, *n* (%)	136 (8.3)	127 (7.9)	9 (20.9)	0.001
Flu-like symptoms, *n* (%)	31 (2.9%)	28 (2.7%)	3 (9.7%)	0.022
Cough, *n* (%)	534 (49.7%)	523 (50.1%)	11 (35.5%)	0.109
White blood cells, ×109/L, median (IQR)	6.2 (4.8–8.5)	6.2 (4.8–8.6)	6.4 (5.0–8.0)	0.881
Neutrophils, ×109/L, median (IQR)	4.3 (2.8–6.5)	4.3 (2.8–6.5)	4.0 (2.7–5.7)	0.542
Lymphocytes, ×109/L, median (IQR)	1130 (780–1630)	1120 (770–1630)	1290 (990–1780)	0.063
Hemoglobin, g/dL, median (IQR)	13.5 (12.1–14.7)	13.5 (12.1–14.7)	13.5 (12.1–14.6)	0.976
Platelets, ×106/L, median (IQR)	219 (172–282)	219 (172–282)	206 (153–288)	0.604
Potassium, mEq/L, median (IQR)	3.6 (3.4–3.9)	3.6 (3.4–3.9)	3.5 (3.3–3.9)	0.532
C-reactive protein, mg/L, median (IQR)	2.9 (1.0–8.2)	2.9 (1.0–8.2)	3.1 (1.3–6.6)	0.910
D-dimer, mg/L, median (IQR)	659 (400–1266)	666 (403–1274)	479 (324–746)	0.014
Ferritin, mg/L, median (IQR)	380 (176–833)	389 (177–838)	243 (159–313)	0.033

Notes: PLWH, people living with HIV; IQR, interquartile range; BMI, body mass index; HCV-Ab, hepatitis C virus antibodies; HBsAg, Hepatitis B surface antigen; PaO_2_/FiO_2_, ratio of arterial oxygen partial pressure to fractional inspired oxygen; NIV, non-invasive ventilation; OTI, oro-tracheal intubation.

**Table 2 jcm-11-01546-t002:** Characteristics of PLWH who were not virologically suppressed.

	Ethnicity	Gender	Age	HIV RNA, Copies/mL	CD4 Count at Admission, Cells/mm^3^	AIDS Defining Disease at Admission	COVID-19 Pneumonia	Comorbidities	PaO_2_/FiO_2_ at Admission, mmHg	Worst PaO_2_/FiO_2_ during FU, mmHg	Oxygen Therapy	Invasiv Ventilation	Death
Patient 1	Black	F	28	222,713	4	Criptococcal meningitis	Yes	No	357	263	yes	IOT	Yes
Patient 2	White	M	53	289	536	Kaposi sarcoma, HIV Encefalopathy	No	COPD	495	495	No	NO	No
Patient 3	White	F	56	4070,603	53	No	No	COPD, HCV chronic hepatitis	462	462	No	No	No
Patient 4	Afro-American	M	48	125	792	No	No	Hypertension, OSAS	386	386	No	No	No
Patient 5	White	F	71	573,680	271	No	Yes	No	338	338	No	No	no
Patient 6	White	M	63	671,810	272	No	Yes	Hypertension, dyslipidemia, CKD	248	195	Yes	No	No
Patient 7	Afro-American	M	48	792	184	No	No	Squamous cell carcinoma of rectum	NA	NA	No	No	no

Abbreviations: COPD, chronic obstructive pulmonary disease; OSAS, obstructive sleep apnea syndrome; CKD, chronic kidney disease; FU, follow up; NA, not available.

## Data Availability

Data supporting reported results can be provided upon request to the authors.

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
