# Peer review of "Characteristics and Outcomes of COVID-19-Related Hospitalization among PLWH"

_jcm, 2022, doi:10.3390/jcm11061546_

Round 1
Reviewer 1 Report
I found the paper interesting, although as the Authors had mentioned, the small sample size of HIV-infected persons makes the obtained results weak. Due to the fact, that the proportion of PLWH in the study group illustrates the proportion of them in the whole italian population, the results could be accepted with all limitations discussed by the Authors.
I suggest only small corrections listed below:
- In the title of the row: „Pao2/FiO2 ratio < 300 at admission, median (IQR)”, „<300 should” be omitted (the title should be: „Pao2/FiO2 ratio at admission, median (IQR)”
- The role of Table 2. is not clear to me, because the (potentially different) results of this subgroup are not presented nor discussed anywhere in the text.
Reviewer 2 Report
I read with great interest your article entitled "Characteristics and outcomes of COVID-19 related hospitalization among PLWH". To date, data on clinical outcomes of PLWH with COVID-19 are conflicting. Poorer outcomes in PLWH may be associated with immunocompromised status or with the presence of other comorbidities, such as cardiovascular diseases and obesity. Nevertheless, otherwise healthy virologically suppressed PLWH doesn't seem to experience a more severe course of SARS-COV-2 infection.
1) In the methods, you define COVID-19 related hospitalization as hospitalization with positive SARS-CoV-2 PCR from nasal/oropharyngeal (NP/OP) swab.
Since your primary endpoint is time to invasive ventilation and death and since the need for invasive ventilation is primarily due to COVID-19 pneumonia, I would reconsider including only patients with pneumonia.
In the results, pneumonia was significantly less frequent in PLWH with COVID-19 compared to non-PLWH patients. This finding raises doubts that hospitalization was really COVID-19 related in these cases (for example patient 2 in Table 2) and doesn't fit with your primary endpoint.
2) Chronic instead of Chronica in the abbreviations of Table 2 and comma is not needed
